# Portfolio Choice for a Resource-Based Sovereign Wealth Fund: An Analysis of Cash Flows

**Knut Anton Mork** [1,*] **, Hanna Marisela Eap** [2] **and Magnus Eskedal Haraldsen** [3]

[1]  Department of Economics, Norwegian University of Science and Technology, Klæbuveien 72,
    N-7491 Trondheim, Norway
[2]  Ernst & Young AS, Dronning Eufemias gate 6, N-0191 Oslo, Norway; hanna.m.eap@no.ey.com
[3]  Danske Bank, Bryggetorget 4, N-0250 Oslo, Norway; mhar@danskebank.com
[*]  Correspondence: knut.anton.mork@ntnu.no; Tel.: +47-9077-5756

**Abstract:** We consider the portfolio choice of a government with a Sovereign Wealth Fund (SWF) when government revenues depend on exhaustible resources, such as oil and gas. The question is whether the SWF portfolio should underweight shares in the resource industry. Some studies have found that these share prices correlate more closely with the overall stock market than the resource price, which would seem to weaken the case for underweighting. However, equity price movements depend not only on changes in expectations of future cash flows, but also on time variation in discount factors. We analyze cash flows directly, rather than trying to disentangle these effects. We have collected cash-flow data for the companies in all of the major industries of the FTSE Global All Cap index, the basis for the strategic index of the Norwegian Government Pension Fund Global. Subsequently, we look at the correlations between each industry's cash flow and the Norwegian government's cash flow from oil and gas. We find a close, statistically significant, and persistent correlation for the oil and gas industry. The correlations for other industries are small and mostly insignificant. We believe that our findings can be used to support proposals for SWFs in countries with significant petroleum revenues to underweight shares in this industry.

**Keywords:** concentration risk; cash flow correlation; Sovereign Wealth Fund

**JEL Classification:** G11; G38

## 1. Introduction

Elementary financial analysis warns against concentration risk. Diversification solves the problem because this kind of risk is idiosyncratic. Accordingly, financial investment portfolios should be diversified. However, many investors face risks outside their portfolios. Diversification for these investors should then consider the full spectrum of the risks that they face, financial risks, as well as background risks. When the background risks are positively (negatively) correlated with some of the financial assets, investment in these assets should then be underweighted (overweighted) relative to a financial portfolio that is perfectly diversified in the absence of background risks. Discussions of this issue on the literature include Bodie et al. (1992), Heaton and Lucas (2000), Viceira (2001), Benzoni et al. (2007), and Cochrane (2014).

Sovereign wealth funds (SWF) are prime examples of financial portfolios that are held by investors with significant background risks. A quick look at the list of significant such SWFs (e.g., Baldwin 2012) reveals that a majority of the leading ones are owned by states with significant resource revenues. Of course, the reason is that the funds have been built up from these revenues. Although temporary

in a long perspective, the revenues usually last for decades. The continued inflow from resource extraction then raises concerns of concentration risks for the SWFs.

The theoretical literature has been rather clear in its recommendation for the SWF to underweight or to avoid completely financial investments in the industry in question. The dynamic optimization analysis by van den Bremer et al. (2016) concluded that, as an example, the Norwegian Government Pension Fund Global (GPFG) should underweight or, indeed, take short positions in oil-industry shares for as long as the country's government continues to receive rent-related revenues from its own oil industry. Henriksen and Kværner (2018) treat unextracted resources as a non-tradeable asset. Like van der Bremer et al., they use the Norwegian GPFG as their prime example. A report from the Norwegian Ministry of Finance (2016a) extends this argument by looking beyond the government's assets to the nation's total capital. Scherer (2019a, 2019b), and Bodie and Brière (2013) are responsible for other contributions to this literature.

Such theoretical arguments must be backed up by empirical facts to be useful for actual decision making. The correlation between financial assets and the asset in the ground should be reliably estimated. However, such estimation is far from trivial. The main problem is that the value of the resources in the ground is unobservable. Markets for such resources may be hard to organize, for example, because of problems of dynamic inconsistency, which can make potential buyers doubt assurances regarding future tax rates or regulations. Nevertheless, if a market is established, the market price might underestimate the social value of the resource, for the same reasons. Finally, because deposits differ widely in regard to geological conditions, the price of one deposit might be a poor indicator for the value of other deposits that have not yet been taken to the market.

One might think that the market value of the extracted resource could work as a proxy for the asset value. It is, after all, the market price of a similar physical product as the one in the ground. In reality, it falls well short of the target, because it ignores the costs of extraction, which are not only significant, but also highly variable, depending on a number of geological, geographical, political, and other factors. Thus, it is hardly surprising that the Norwegian Ministry of Finance (2018) finds that the correlation between oil prices and oil-company stock prices is too low to warrant the underweighting of oil and gas shares in the GPFG. In fact, they find that the returns to oil and gas shares are more closely correlated with the overall equity market than with the price of oil.

NBIM, Norges Bank Investment Management (2017a), which is the agency managing the investment of the GPFG, has looked closer at these correlations. They seek to separate stock price movements driven by changes in discount rates from those following from revisions of expectations of future cash flow while using the arguments of Campbell and Shiller (1988) and the method of Campbell (1991). They find empirical support for the claim that the close correlation between oil and gas stock returns and the overall stock market is mainly driven by variations in discount rates, which should be common across all industries. After separating out this effect, they find that the component reflecting revisions of cash-flow expectations is much more closely correlated with the world price of oil.

While being highly suggestive, this contribution does not settle the issue. As the price of oil is a highly imperfect proxy for the value of oil in the ground, it compares apples and oranges. Furthermore, Campbell's method is fraught with sampling as well as approximation errors from its reliance on linear approximations and key identifying assumptions.

The main contribution of this paper is to circumvent these problems by instead looking directly at the correlation between oil and gas companies' cash flows and the Norwegian government's cash flow from oil and gas activities[1]. Furthermore, we compare this correlation with the corresponding correlations for the other major industries represented in the global stock market. By directly studying cash flows, we avoid the complications that are caused by time-varying discount factors. We also avoid the possible biases from linear approximations and special identifying assumption. On the other

---

[1]　　This idea was introduced to us by an op-ed article in a Norwegian business daily by Hoddevik (2018).

hand, our analysis is subject to expectation errors by studying realized rather than expected future cash flows. If expectations are rational, this should not matter. Additionally, even if they are biased, our results should give a reliable guide to the relationship between the company asset prices and the imputed asset price of the government's resource holding, only provided that the biases tend in the same direction. At the very least, we offer an alternative perspective on an issue for which a perfect method of analysis is simply not available.

Pettit and Westerfield (1972) used cash flow data in an early extension of the single-factor capital asset market model (CAPM) by Sharpe (1964) and Lintner (1965). More recently, Cohen et al. (2009) argued that noise in high-frequency return data might distort tests of market efficiency and used cash-flow data to derive results to support their claim that the CAPM provides a satisfactory description of price-level data for buy-and-hold investors. In a similar vein, Campbell et al. (2010) compared cash-flow data to Campbell (1991) model in a study of growth stocks and value stocks.

It is worth emphasizing that the concept of cash flow in these studies, including Campbell (1991), refers to dividends rather than the underlying company cash flows. The focus on dividends naturally makes sense from a stock investors point of view. However, fundamentally, it is the cash flow that is generated by the companies that form the basis for stock values. Furthermore, the Norwegian government's petroleum revenues behave much more like company cash flows than dividends. Except for the small portion coming from Equinor (formerly Statoil) dividends, these revenues are roughly equally split between the special 78% tax rate on oil company profits and the cash flow from the government's direct financial investment in oil and gas fields. Consequently, we find it natural to study the correlation between these revenues and the cash flow that is generated by listed companies in the respective sectors.

A limitation of our study lies implicitly in the relatively young age of the Norwegian GPFG. Although its establishment was enacted by Parliament in 1990, the first deposit into the fund, corresponding to about USD 300 million, was not made until 1996; and, the first equity investments were not made until 1998. Furthermore, the fund's current equity index, which is based on the FTSE Global All Cap index, was not launched until 2003, which forms the start of our sample. A look at the correlation between cash flows of oil and gas companies and the corresponding ones of the Norwegian government further back in time would nevertheless have been interesting. However, after serious consideration of this issue, we had to conclude that problems of data availability would make the results of any such effort uninformative.[2]

Using quarterly data could have increased the number of observations in our study, although not the length in calendar time. However, data for the Norwegian government's net petroleum cash flow (GPCF) are only available in a meaningful way on the annual frequency. A quarterly breakdown would be dominated by irrelevant details concerning, for example, the timing of tax payments rather than the time profile of the actual earnings. This limits our sample to the 16 annual observations (15 first differences) between 2003 (2004) and 2018.

These facts obviously limit the power of our statistical tests. On that background, we find it remarkable that we nevertheless find highly significant results.

We find that the cash flows of oil and gas companies are very closely correlated with the Norwegian government's cash flow from oil and gas activities. The corresponding correlations for the other industries are either much weaker or negative. Within the oil and gas industry, we find the strongest correlation for the subsector of integrated oil companies. We also find a significant, but much weaker,

---

[2]   As can be seen from Figure 1 below, the government revenues were much smaller before the turn of the century and thus likely more influenced by idiosyncratic factors. Furthermore, oil and gas was not identified as a sector by the suppliers of equity indices until the mid-2000s. Indeed, some of the data for the early part of our sample had to be assembled manually. For data further back in time, we thus would have had to rely on just a few major companies, which would be likely to bias our results.

correlation for the subsector of companies with only upstream operations in oil and gas. We find no significant correlations with the government's cash flow for the other oil and gas subsectors.

We believe these results to be supportive of the argument that the shares of integrated oil companies and upstream companies should be underweighted in the strategic indices of oil-rich countries' SWFs, such as the Norwegian GPFG. Our results are only weakly supportive of the Norwegian government's recent decision[3] to only underweight the shares of upstream companies. The explanation given for that decision was that the integrated oil and gas companies do more than resource extraction. However, we believe the Norwegian government ignored the dominant actors in the global upstream petroleum industry by excluding this subgroup.

The organization of the rest of the paper is as follows. Section 2 presents the GPFG. Section 3 presents the data for the Norwegian government's petroleum cash flow, and Section 4 displays the data for the cash flows of the global corporations that are relevant for the GPFG. Section 5 presents the main results of our analysis. Section 6 offers some robustness checks, and Section 7 concludes.

## 2. The Government Pension Fund Global

The GPFG is the Norwegian government's vehicle for investing the extraordinary proceeds from the extraction of non-renewable petroleum resources and, thus, preserving this wealth for future generations. The government's cash flow from oil and gas activity, to be further described in the next section, is the fund's only source of deposits. By statute, all of this cash flow is deposited each year into the fund. Norges Bank Investment Management (NBIM) manages the fund, a division of the central bank. An Act of Parliament stipulates that the Government, in practice the Ministry of Finance, acts as the fund's owner and formulates its investment strategy, subject to Parliamentary approval.

A Fiscal Rule, which was adopted by Parliament in 2001[4], allows for an annual draw from the fund corresponding to its expected real return. At the outset, the expected real rate of return was estimated as 4%. This rate was officially lowered to 3% in the 2018 budget after the ensuing experience of low riskless rates and a lengthy public debate. Although the rule allows for considerable flexibility, especially in response to the domestic business cycle, it has mainly been respected[5]. Even so, more than 15% of all government spending is currently financed from this source because of its fast growth to a total AMU exceeding USD 1 trillion.

The fund is invested in equity, fixed income, and real estate in the global economy, with a certain overweight of European assets. Securities that are issued by Norwegian entities or in Norwegian kroner are excluded from the fund's universe. Index replication governs most of the investments, although a small portion is actively managed[6]. The current strategic benchmark, issued by the Ministry of Finance, specifies an equity share of up to 70% and the rest in fixed income.[7] Investment in unlisted real estate can, at most, account for 7% of the total AUM. However, in index terms, such investments are considered part of the equity share and evaluated accordingly.

Bloomberg provides the fund's fixed-income benchmark index. It consists of the following three subindices: Global Treasury GDP, Global Inflation-linked, and Global Aggregate. It is made up of 70% government bonds in 21 different currencies, and 30% corporate bonds in seven different currencies. The government bonds are weighted according to each country's GDP, and corporate bonds are weighted based on each company's outstanding debt.

The equity benchmark is based on the FTSE Global All Cap Index (GEISAC). It is market weighted and includes large, mid, and small cap stocks in both Developed and Emerging markets. The GEISAC index contains shares of around 8000 companies in 49 different countries. It was launched by

---

[3]　Norwegian Ministry of Finance (2019).
[4]　Norwegian Ministry of Finance (2001).
[5]　Norwegian Ministry of Finance (2015).
[6]　A further discussion can be found in Chambers et al. (2012).
[7]　Norwegian Ministry of Finance (2016b).

FTSE Russell in 2003 and was developed to be used for index tracking funds, derivatives, and as a performance benchmark for funds, such as the GPFG.

The GPFG actually uses a modified version of the GEISAC, as stipulated by the Ministry of Finance. The greatest modification is the overweighting of European corporations and the exclusion of Norwegian ones. Shares by non-Norwegian European corporations are weighted at 2.5 times their actual market cap. For U.S. and Canadian corporations, the corresponding weights are unity; and, for the remaining developed and emerging markets, the weights are 1.5. Furthermore, the GPFG invests in the equity markets of twenty countries not included in the GEISAC index, including local Chinese equity (China A), Croatia, Saudi Arabia, and Morocco. Finally, some corporations are excluded for ethical or environmental reasons, such as tobacco, coal, and tar-sand companies.

The GPFG is allowed to deviate from the strategic benchmark with a maximum deviation of an expected relative volatility of 1.25 percentage points in response to sudden market movements. The expected realized volatility is a measure of how much the return on the GPFG is expected to deviate from the benchmark index return in a normal year (Norges Bank Investment Management 2018). The NBIM is required to rebalance the equity allocation whenever the equity share significantly deviates from the strategic benchmark index.

Our main interest in this paper is the fund's investment in oil and gas companies. Table 1 displays the overall industry allocation in the FTSE GEISAC index as well as the actual holdings of the GPFG at the end of 2018. The industry classification follows the FTSE Russel Industry Classification Benchmark (ICB). The Oil and Gas industry makes up 5.9% of the fund as well as the FTSE index. Although not an overly large sector when compared, e.g., to financials with a share of 23.7%, it is of special interest because investment in this sector might give rise to a concentration risk for the Norwegian economy.

**Table 1.** The company count and industry weights for companies in each Industry Classification Benchmark (ICB) industry in the Government Pension Fund Global (GPFG) and FTSE Global All Cap Index (GEISAC), as well as their market value in the GPFG, as of Dec. 31, 2018. Sources: Norges Bank Investment Management (2017b) and FTSE Russell (2019).

| | **GPFG Equity Holdings** | | | | **FTSE Global All Cap** | |
|---|---|---|---|---|---|---|
| **Industry No.** | **Industry** | **Count** | **NOK mn** | **Weight %** | **Count** | **Weight %** |
| 0001 | Oil and Gas | 341 | 320,756 | 5.9 | 320 | 5.9 |
| 1000 | Basic Materials | 659 | 271,304 | 5.0 | 614 | 4.6 |
| 2000 | Industrials | 1966 | 708,762 | 12.9 | 1651 | 13.4 |
| 3000 | Consumer Goods | 1204 | 653,764 | 11.9 | 1009 | 11.0 |
| 4000 | Health Care | 723 | 626,847 | 11.4 | 544 | 11.2 |
| 5000 | Consumer Services | 1204 | 589,709 | 10.8 | 1008 | 11.5 |
| 6000 | Telecommunications | 130 | 163,344 | 3.0 | 129 | 2.8 |
| 7000 | Utilities | 252 | 155,333 | 2.8 | 286 | 3.3 |
| 8000 | Financials | 1859 | 1,299,103 | 23.7 | 1659 | 21.9 |
| 9000 | Technology | 809 | 689,838 | 12.6 | 644 | 14.4 |
| | Total | 9158 | 5,478,760 | 100.0 | 7864 | 100.0 |

However, we should note that this industry includes companies that are not involved in petroleum extraction, such as pipeline companies and renewable-energy companies. For this reason, we also look at a further breakdown into subsectors as displayed in Table 2. As seen there, Integrated Oil and Gas dominates the Oil and Gas industry, with a share of almost two-thirds. This subsector contains the oil majors, such as Royal Dutch Shell, ExxonMobil, BP, and Chevron. These companies are involved in all parts of petroleum production, from exploration and drilling to refining and distribution. They may even be involved in other energy forms, such as renewables. However, we will also be interested in the second-largest subsector, Exploration and Production, which makes up one-fifth of the Oil and Gas industry. Here, we find upstream activities, like exploration, drilling, production, refining, and supply of oil and gas products (FTSE Russell 2019). *Ex ante* one might expect these activities to be most closely correlated with the activities generating the Norwegian government's petroleum revenues.

**Table 2.** The company count and industry weights for companies in the Oil and Gas subsectors in the GPFG and FTSE GEISAC, as well as their market value in the GPFG, as of 31 December 2018. Sources as in Table 1.

| | | GPFG Benchmark Index | | | FTSE Global All Cap | |
|---|---|---|---|---|---|---|
| Subsector No. | Industry | Count | NOK mn | Wt % | Count | Weight % |
| 0533 | Exploration and Production | 134 | 70,276 | 20.52 | 138 | 25.08 |
| 0537 | Integrated Oil and Gas | 61 | 223,066 | 65.13 | 62 | 57.88 |
| 0573 | Oil Equipment and Services | 73 | 20,227 | 5.91 | 78 | 6.98 |
| 0577 | Pipelines | 16 | 22,546 | 6.58 | 16 | 8.64 |
| 0583 | Renew. Energy Equipment | 23 | 6287 | 1.84 | 23 | 1.39 |
| 0587 | Alternative Fuels | 3 | 87 | 0.03 | 3 | 0.03 |
| | Total | 310 | 342,489 | 100.00 | 320 | 100.00 |

The Pipelines subsector was established in 2006 when the industry classification benchmark ICB scheme was introduced. The subsectors Renewable Energy Equipment and Alternative Fuels were created in 2009. Thus, we do not have data for these subsectors from the earlier years.

## 3. The Norwegian Government's Petroleum Cash Flow (GPCF)

The Norwegian government started to collect revenue from oil activity on its continental shelf after the first discovery of oil in December of 1969. Figure 1 shows the time series of these revenues in NOK 2019, converted to U.S. dollars. Although the first revenues started to trickle in shortly after oil was first discovered, they remained modest until a huge jump around the turn of the century. The annual revenues mainly stayed above USD 30 billion for our 2003–2018 sample.

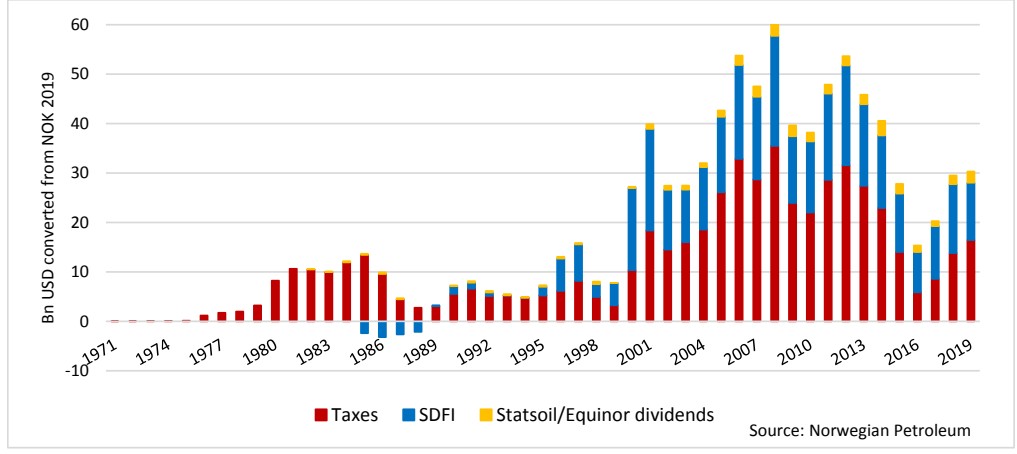

**Figure 1.** The Norwegian government's net cash flow from petroleum activities.

Figure 1 also presents the breakdown of this revenue stream into its three main components:

- Oil company taxes, mainly the special corporate income tax for this sector, as explained below. Environmental taxes, area fees, and other fees, are also included, though their contribution is minor.
- The net cash flow from the government's direct participation as a financial partner in oil field development and operation via the State Direct Financial Investment (SDFI) program.
- Dividends from Equinor (formerly Statoil).

For most of the years, the greatest share has come from oil-company taxes, mainly in the form of a special corporate income tax, as defined in the Petroleum Taxation Act of 1975. Oil companies do not

pay for their production licenses. However, in addition to the normal corporate tax rate, currently 22%, profits from offshore operations are charged an additional tax that raises the marginal rate on such profits to 78%. This scheme is intended to mimic a rent tax, such that, in an approximate sense, the government receives the entire resource rent. A deductible amount is added to take care of the extensive margin, that is, to make sure that projects that are socially beneficial are also profitable after taxes. In addition, a reimbursement system for exploration costs was introduced in 2005, by which the government reimburses the company for 78% of exploration activities, even if the company has no taxable profits.

The SDFI is the Norwegian government's direct ownership shares in oil and gas fields, onshore facilities, and pipelines. This ownership share, which varies from field to field, is jointly determined with the awarding of production licenses. At the end of 2018, the SDFI portfolio consisted of shares in 38 fields, four of which are in the development phase, and 15 terminals and pipelines, representing approximately one third of the country's total petroleum reserves. Via this route, the government only participates as a financial investor, contributing to investment expenditures as well as receiving its share of revenues. Since 2001, Petoro, a 100% government owned corporation, has acted as the government's agent for these investments.

Until 2001, the government also owned all of the shares of Statoil (now Equinor). The government share has been reduced to 67% since listing on the Oslo Stock Exchange that year. Dividends from this company have always only represented a small share of the government's overall petroleum revenues, as shown in Figure 1.

We converted each year's revenues from Norwegian kroner into U.S. dollars, using the average exchange rate for that year, in order to make our data for the government's petroleum revenues comparable to the cash flows of global corporations. This makes the data used in our analysis slightly different from the ones in Figure 1, which we find to be better suited for illustrative purposes.

## 4. Industry Operating Cash Flows

One of the main contributions of this paper is the data that we have collected for the cash flows of companies in the global stock market, sorted by industry. For this purpose, we have combined data from several databases. We define operating cash flow as the sum of net revenue and non-cash expenses, net of changes in working capital. Operating cash flows can be reinvested in the company through fixed asset investments, used to reduce debt, or be directly paid out to owners as dividends. Although shareholders do not receive this cash flow, it forms the basis for how the companies are valued in the stock market.

Company cash flows are reported in local currency, 41 individual currencies in all. We converted these amounts to U.S. dollars while using annual averages of daily exchange rates. We used exchange rates from the FRED database for the 23 currencies available there and the rest from Macrobond. Apart from this conversion, we analyze all of our cash flows in nominal terms.

The cash-flow data for each company were taken from the S & P database Capital IQ, which we accessed via the Wharton Research Data Services. Each company was identified by means of CUSIP codes for North American companies and SEDOL codes for the rest. We obtained those codes from FTSE Russel for the companies that were included in the FTSE GEISAC index. This way, we obtained annual USD cash-flow data for 86,622 company-years out of the 121,581 data points from the FTSE, or 71.2% of the total. Table 3 shows the numbers of companies in our data set along with their aggregate weights in the GEISAC index.

In 88 instances, we found multiple observations for a company's operating cash flow within the same year. A number of Asian companies have reported annual figures for different time periods than January–December, such as March, February, or June. We then chose one of these for our annual observation. As an example, the figure that we use for operating cash flow for Keyence Corp in 2017 is the one reported in March 2018. For a few cases, the companies have switched from March to December as the end-of-period, resulting in two separate figures for the same calendar year. In these cases, we

have used the figures that were reported at the same time of the year as in the following years so as to obtain consistent time series. Typically, these companies switched from June to March reporting.

**Table 3.** The number of companies for which we were able to find cash flow data, as well as their aggregate weights, in all industries and the Oil and Gas industry. The corresponding numbers for the companies in the FTSE GEISAC are shown for comparison.

| | All Industries | | | | OIL and GAS | | | |
| | Our Data | | GEISAC | | Our Data | | GEISAC | |
| Year | Count | Weight | Count | Weight | Count | Weight | Count | Weight |
|---|---|---|---|---|---|---|---|---|
| 2003 | 4053 | 76.1% | 6959 | 100% | 130 | 6.0% | 206 | 6.6% |
| 2004 | 4806 | 79.1% | 7595 | 100% | 157 | 6.6% | 231 | 7.3% |
| 2005 | 5293 | 79.8% | 8080 | 100% | 199 | 7.7% | 286 | 8.5% |
| 2006 | 5436 | 80.5% | 8116 | 100% | 246 | 8.2% | 334 | 8.8% |
| 2007 | 5743 | 84.7% | 7920 | 100% | 268 | 9.8% | 343 | 10.4% |
| 2008 | 5735 | 87.7% | 7756 | 100% | 284 | 10.0% | 366 | 10.6% |
| 2009 | 5572 | 87.5% | 7304 | 100% | 316 | 9.8% | 391 | 10.4% |
| 2010 | 5615 | 87.4% | 7301 | 100% | 322 | 9.7% | 393 | 10.4% |
| 2011 | 5785 | 89.1% | 7408 | 100% | 340 | 10.5% | 414 | 11.0% |
| 2012 | 5469 | 86.6% | 7197 | 100% | 328 | 9.1% | 395 | 9.7% |
| 2013 | 5535 | 87.5% | 7241 | 100% | 327 | 8.5% | 384 | 8.9% |
| 2014 | 5838 | 87.8% | 7580 | 100% | 352 | 7.1% | 418 | 7.3% |
| 2015 | 6037 | 88.9% | 7747 | 100% | 335 | 5.7% | 381 | 5.8% |
| 2016 | 6088 | 89.8% | 7725 | 100% | 306 | 6.8% | 336 | 7.0% |
| 2017 | 6231 | 90.0% | 7788 | 100% | 303 | 5.8% | 319 | 6.0% |
| 2018 | 3386 | 68.5% | 7864 | 100% | 219 | 5.2% | 320 | 5.9% |
| Average | 5414 | 84.4% | 7599 | 100% | 277 | 7.9% | 345 | 8.4% |

The importance of an industry's cash flow to the GPFG depends on that industry's weight in the fund's index. We weighted each company's cash flow by its industry's weight in the GPFG's index to capture this importance, and to correct for the missing data, the extent of which varies by industry, according to the formula

$$\text{Weighted } C_{jt} = \sum_{i=1}^{N_j} w_{jt} C_{ijt},$$

where $C_{ijt}$ is the cash flow of company $i$ in industry $j$ for year $t$, $N_j$ is the number of companies within industry $j$, and $w_{jt}$ is the index weight of industry $j$ at time $t$. The industry weights, which vary slightly from year to year, are shown in Table 4.

**Table 4.** The weights of ICB industries in the Government Pension Fund Global over time. Source: Norges Bank Investment Management (2019).

| Year | Oil and Gas | Basic Mat. | Industrials | Cons. Goods | Health Care | Cons. Serv- | Telecom | Utilities | Financials | Technology |
|---|---|---|---|---|---|---|---|---|---|---|
| 2003 | 7.9% | 4.8% | 10.3% | 10.8% | 10.6% | 11.2% | 6.9% | 3.0% | 25.3% | 9.2% |
| 2004 | 8.3% | 4.6% | 10.8% | 9.8% | 9.9% | 11.6% | 7.0% | 3.4% | 26.7% | 7.9% |
| 2005 | 9.4% | 5.5% | 10.8% | 9.9% | 9.6% | 10.3% | 5.1% | 3.7% | 27.1% | 8.7% |
| 2006 | 8.0% | 5.6% | 11.1% | 10.7% | 8.0% | 9.6% | 4.7% | 4.8% | 29.3% | 8.1% |
| 2007 | 10.1% | 7.5% | 12.4% | 11.8% | 7.7% | 8.7% | 5.3% | 5.2% | 23.2% | 8.3% |
| 2008 | 11.0% | 6.3% | 11.4% | 12.1% | 10.3% | 8.6% | 6.1% | 6.0% | 21.1% | 7.1% |
| 2009 | 10.8% | 8.0% | 12.0% | 11.3% | 8.6% | 8.3% | 5.1% | 4.8% | 22.8% | 8.3% |
| 2010 | 10.8% | 9.1% | 13.7% | 11.6% | 7.7% | 8.5% | 4.5% | 4.7% | 21.5% | 7.9% |
| 2011 | 11.6% | 7.8% | 13.1% | 12.7% | 9.5% | 9.0% | 4.4% | 4.3% | 19.8% | 7.9% |
| 2012 | 9.9% | 7.5% | 13.1% | 13.6% | 8.7% | 9.3% | 3.9% | 3.8% | 23.0% | 7.3% |
| 2013 | 8.3% | 6.3% | 14.3% | 13.9% | 8.7% | 10.2% | 3.8% | 3.4% | 23.6% | 7.4% |
| 2014 | 6.9% | 5.7% | 13.6% | 13.8% | 9.6% | 10.4% | 3.3% | 3.7% | 24.5% | 8.4% |
| 2015 | 5.4% | 5.1% | 13.5% | 14.4% | 10.7% | 10.9% | 3.4% | 3.2% | 24.5% | 9.0% |
| 2016 | 6.4% | 5.6% | 14.0% | 13.6% | 10.1% | 10.2% | 3.2% | 3.1% | 24.3% | 9.4% |
| 2017 | 5.6% | 6.0% | 14.3% | 13.5% | 9.8% | 10.1% | 2.8% | 2.6% | 24.3% | 11.1% |
| 2018 | 5.9% | 5.0% | 12.9% | 11.9% | 11.4% | 10.8% | 3.0% | 2.8% | 23.7% | 12.6% |

Our data set has two main weaknesses beyond the limited number of years for which we have observations.

1. We find cash flow data based on the FTSE GEISAC constituents, not the GPFG constituents, due to the lack of company codes from the GPFG holding reports. Thus, all companies in the GPFG, not part of the FTSE GEISAC, are left out of our analysis.
2. We were not able to find all companies of the FTSE GEISAC index in the Capital IQ database. Furthermore, the observations of operating cash flow were not necessarily available for some of the companies that we did find. To make up for this problem, we manually collected cash flows for a number of companies from their respective annual reports. In the interest of efficiency, when we did this, we sorted the companies by market capitalization and made sure our resulting data set included all of the 300 largest companies for each year.

Table 5 presents our estimates of the annual cash flows in 2003–2018 for each industry as well as the Norwegian government's petroleum cash flow.

**Table 5.** Operating cash flows for each ICB industry 2003–2018, weighted by the GPFG industry weights. The last column shows the Norwegian government's petroleum cash flow. USD billions.

| Year | Oil and Gas | Basic Mat. | Indus-trials | Cons. Goods | Health Care | Cons. Serv. | Telecom | Utilities | Financials | Technology | GPCF |
|---|---|---|---|---|---|---|---|---|---|---|---|
| 2003 | 20.5 | 6.3 | 31.6 | 31.3 | 13.8 | 21.9 | 19.7 | 4.6 | 123.4 | 13.3 | 24.5 |
| 2004 | 27.6 | 8.7 | 37.0 | 31.9 | 15.1 | 27.0 | 22.1 | 5.8 | 101.4 | 14.8 | 30.2 |
| 2005 | 40.7 | 12.5 | 42.9 | 30.8 | 16.0 | 24.7 | 17.1 | 6.9 | 38.4 | 18.4 | 42.8 |
| 2006 | 41.8 | 14.7 | 47.1 | 37.6 | 14.0 | 28.9 | 16.5 | 10.4 | 79.7 | 18.0 | 55.4 |
| 2007 | 57.6 | 24.2 | 68.0 | 49.9 | 16.2 | 31.7 | 21.1 | 12.4 | 163.4 | 22.4 | 54.0 |
| 2008 | 76.3 | 21.5 | 59.5 | 37.9 | 22.4 | 31.8 | 25.4 | 14.5 | 231.3 | 18.7 | 73.8 |
| 2009 | 57.1 | 26.6 | 62.3 | 62.6 | 19.8 | 32.7 | 21.5 | 15.6 | 199.8 | 22.8 | 44.5 |
| 2010 | 70.6 | 36.5 | 70.8 | 59.1 | 18.5 | 36.9 | 20.3 | 15.3 | 232.3 | 26.0 | 45.6 |
| 2011 | 90.9 | 37.3 | 65.0 | 57.1 | 24.2 | 41.9 | 20.5 | 12.9 | 268.1 | 27.4 | 62.6 |
| 2012 | 81.3 | 31.1 | 67.1 | 75.4 | 21.4 | 42.5 | 17.6 | 11.5 | 248.0 | 25.0 | 68.0 |
| 2013 | 73.3 | 25.1 | 80.5 | 84.7 | 21.8 | 48.7 | 16.7 | 11.4 | 252.8 | 27.2 | 58.7 |
| 2014 | 62.5 | 23.3 | 80.3 | 81.1 | 27.0 | 53.2 | 13.7 | 12.8 | 250.5 | 35.1 | 49.5 |
| 2015 | 35.6 | 19.3 | 82.3 | 94.4 | 31.6 | 56.7 | 13.7 | 11.4 | 347.0 | 40.0 | 27.1 |
| 2016 | 33.9 | 20.8 | 90.6 | 93.6 | 32.3 | 56.3 | 13.8 | 10.4 | 354.0 | 45.6 | 14.9 |
| 2017 | 36.9 | 24.3 | 89.2 | 98.1 | 33.9 | 61.0 | 12.1 | 8.3 | 336.0 | 61.0 | 20.3 |
| 2018 | 36.2 | 13.4 | 54.8 | 50.7 | 34.4 | 47.3 | 8.4 | 5.7 | 189.1 | 75.3 | 30.9 |

## 5. Analysis

Our analysis starts from the standard pricing formula for an asset or asset portfolio indexed *i* as the expected present value of the present and future cash flow generated by the asset[8]:

$$V_{it} = \mathbb{E}_t \sum_{s=0}^{\infty} \beta^s C_{i,t+s}.$$

Time variations in this value may occur either because of unexpected changes in the cash flow or because of changes in the discount factor *β*, as pointed out by Campbell and Shiller (1988). For the case where all of the relevant asset values are observable, Campbell (1991) has designed a method for statistically separating the two forces of variation. However, in our application, observations are missing for one key asset of interest, namely, the government's stock of unextracted resources. We do, on the other hand, have data for realized cash flows. In principle, they could be used to construct implied asset values ex post. However, that would have required data for cash flows much farther into the future than has as yet been realized. Our alternative is then to analyze the cash flows themselves. The cross-sector correlations of these cash flows should not be distorted by time variation in the discount factors as long as these changes are uniform across sectors. This way, we sidestep the complications for which Campbell designed his separation identification method.

---

[8] This formula properly applies to dividends, not operating cash flows. When applied to operating cash flows, as we do, it involves some double counting to the extent that operating cash flows have been reinvested in order to generate future cash flows. We do not believe that this issue biases our results significantly, however.

Statistically speaking, suppose that $(x_1, y_1)$ and $(x_2, y_2)$ are independent draws from the same bivariate distribution according to which $corr(x_j, y_j) = \rho$. Then, for given $\beta$, $corr(x_1 + \beta x_2, y + \beta y_2) = \rho$ as well. Indeed, any such linear combinations of draws will have the same correlation. Applied to the above valuation formula, this means that, for given discount factors, the correlation between the cash flows should carry over to the asset values. This insight motivates our investigation of the correlations between the government's petroleum cash flow on the one hand and the corporate cash flows of the respective global industries on the other.

Figure 2 displays the time series for each industry's cash flow along with the Norwegian government's cash flow from petroleum activities, all being normalized to 100 in 2003. These graphs suggest a very strong relationship between government's petroleum cash flow and the cash flow for the Oil and Gas industry. A relationship, albeit weaker, appears to also be present for Basic Materials and Utilities. For the other industries, the relationships seem weak or non-existent.

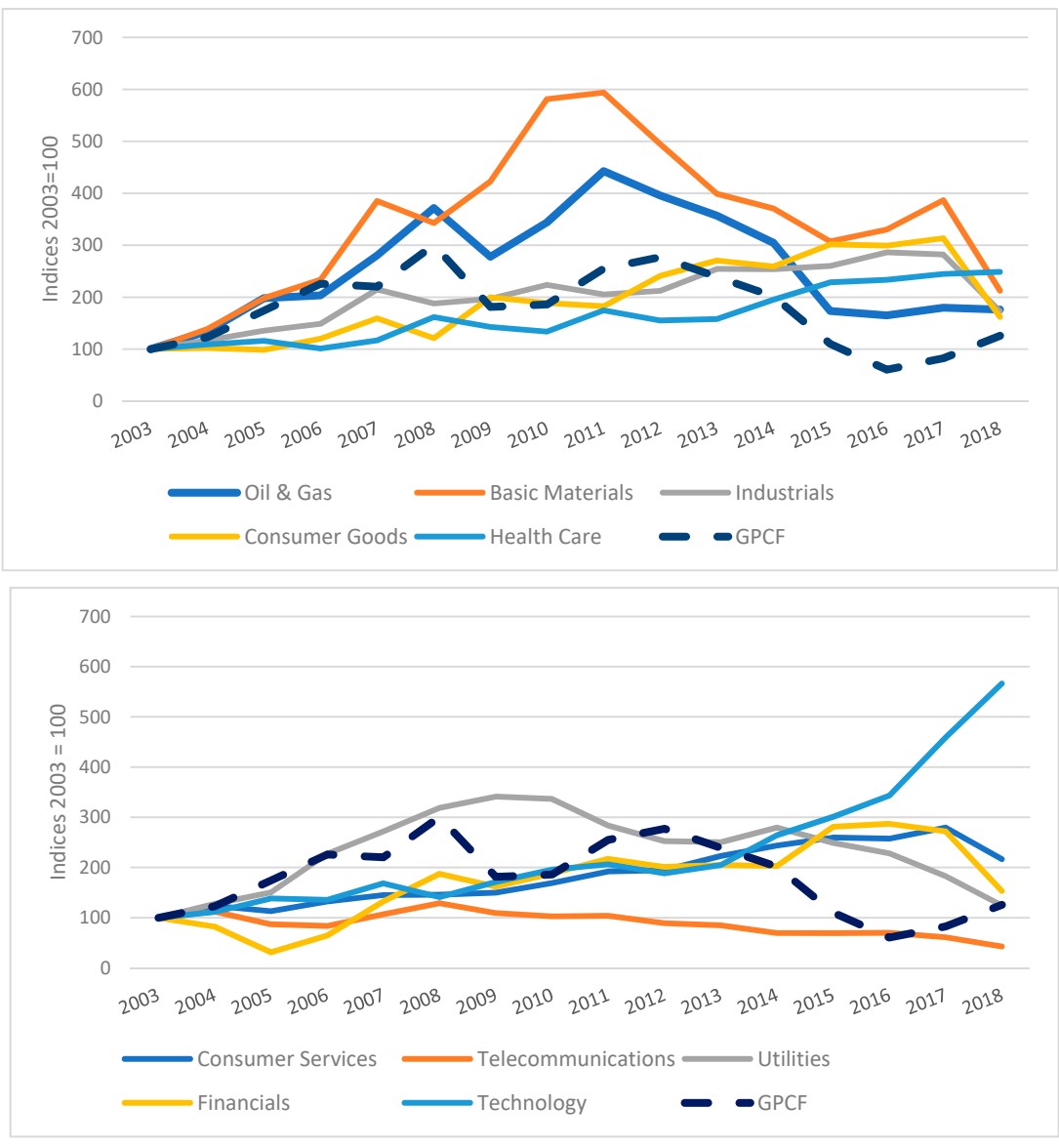

**Figure 2.** Time series of weighted operating cash flows for each of the ICB industries as well as the Norwegian government's cash flow from petroleum activities (GPCF), indexed, 2003–2018.

However, the apparent correlations in these graphs may be deceiving because of the presence of unit roots, which are known to give rise to spurious correlations. Thus, we start our formal analysis by

testing for the presence of unit roots. Table 6, which shows the results of the Dickey–Fuller tests for all the relevant series, indicates that the unit-root hypothesis cannot be rejected for any of the series. We realize that these tests have low power because of the limited length of the sample. However, unit roots should be expected *ex ante* for these series, because, as nominal cash-flow series, they are influenced by trend inflation (however weak in this period) as well as global economic growth. Moreover, for series with near-unit roots, inference in small samples is usually more reliable when conducted on first differences than on the original series. Thus, we base our analysis on the differenced series.

**Table 6.** *p* values from Dickey-Fuller tests of the GPCF and each of the weighted operating cash flow series for the major ICB industries.

| Series | I(1) | I(2) |
|---|---|---|
| GPCF | 0.427 | 0.021 |
| Oil and Gas | 0.443 | 0.030 |
| Basic Materials | 0.393 | 0.141 |
| Industrials | 0.264 | 0.173 |
| Consumer Goods | 0.483 | 0.015 |
| Health Care | 0.914 | 0.001 |
| Consumer Services | 0.576 | 0.017 |
| Telecommunications | 0.916 | 0.008 |
| Utilities | 0.596 | 0.672 |
| Financials | 0.579 | 0.302 |
| Technology | 1.000 | 0.574 |
| All ICB industries ex Oil and Gas | 0.553 | 0.544 |
| GEISAC | 0.500 | 0.510 |

For the tests of a second unit roots, about half of the series showed rejection on the 5% level. Again, when considering the low power of these tests, we believe that they support our decision to work with first differences.

The series for the subsectors of the Oil and Gas showed similar results, as presented in Table 7.

**Table 7.** *p* values from Dickey–Fuller tests of the weighted operating cash flow of the Oil and Gas subsectors.

| Series | I(1) | I(2) |
|---|---|---|
| Exploration and Production | 0.427 | 0.021 |
| Integrated Oil and Gas | 0.443 | 0.030 |
| Oil Equipment and Services | 0.393 | 0.141 |
| Pipelines | 0.264 | 0.173 |
| Renewable Energy Equipment | 0.483 | 0.015 |
| Alternative Fuels | 0.914 | 0.001 |
| Oil and Gas ex Exploration and Production | 0.553 | 0.544 |

Our main results are the correlations, in first differences, between the government's petroleum cash flow on the one hand, and on the other hand the cash flow for the companies in each of the major industries, respectively. Table 8 presents these results.

**Table 8.** Correlations between the GPFG-industry weighted operating cash flow of each industry and the GPCF. All correlations on first-difference data 2004–2018.

| Industry | Correlation | *t*-Value |
|---|---|---|
| Oil and Gas | 0.79 | 4.66 *** |
| Basic Materials | −0.08 | −0.27 |
| Industrials | −0.34 | −1.32 |
| Consumer Goods | −0.50 | −2.10 * |
| Health Care | 0.19 | 0.70 |
| Consumer Services | −0.19 | −0.68 |
| Telecommunications | 0.05 | 0.17 |
| Utilities | 0.20 | 0.75 |
| Financials | −0.14 | −0.49 |
| Technology | −0.19 | −0.68 |
| All industries | −0.12 | −0.44 |
| All industries ex Oil and Gas | −0.25 | −0.93 |

*** $p < 0.01$, * $p < 0.1$.

Although the shortness of the sample warns against firm conclusions, we find the overall pattern that emerges from Table 8 quite convincing. Only one industry, Oil and Gas, shows a significant, positive correlation with the government's petroleum cash flow. Not only that, this correlation coefficient of 0.8 is four times as high as the second-largest one, which is Telecommunications at 0.2. In fact, all of the other correlations seem trivially low, except for the marginally significant negative correlation for Consumer Goods. The correlation for the total of all industries is insignificantly negative whether or not the Oil and Gas industry is included.

Moreover, we find the result for Oil and Gas reasonable when considering the rules governing the Norwegian government's cash flow. Two parts of it, oil company income taxes and Equinor dividends, are based on oil company profits, and the third part, the net SDIF cash flow, is directly proportional to the cash flow of the government's partners in the oil fields in question.

We believe these results can be interpreted as strong support for underweighting of companies in the Oil and Gas industry in the strategic index for SWFs of oil-rich governments, such as the Norwegian GPFG. However, as the government's cash flow comes mainly from oil companies' upstream activities, it could be argued that the underweighting argument only applies to that part of the Oil and Gas industry.

Table 9 shows the correlations between the Norwegian government's petroleum cash flow and that of the respective Oil and Gas subsectors, again in first differences. We used the FTSE GEISAC weights because we were unable to obtain the weights for these subsectors in the GPFG. The Exploration and Production subsector indeed shows a correlation of 0.5, which is marginally significant at the 5% level. However, the sector labeled Integrated Oil and Gas has a correlation coefficient of almost 0.9, which is significant at the 0.1% level. The coefficients for the remaining subsectors are small and insignificant, with the possible exception of the Renewable Energy Equipment sector, which shows a negative correlation coefficient of 0.59. However, for this sector, we only had data from 2010 to 2018.

We believe that the close correlation for the Integrated Oil and Gas sector is due to the fact that this sector includes the main global players in the oil and gas industry and, thus, reflects the main global trends of the industry. This sector made up about 60% of the capitalization of the entire Oil and Gas industry for our sample period, as shown in Table 2. The members of the Exploration and Production sector tend to be smaller, which would make their aggregate cash flow more influenced by idiosyncratic factors. As further support of that view, we note that the correlation for the entire Oil and Gas industry, except the Exploration and Production subsector, is just as high as the one for Integrated Oil and Gas.

**Table 9.** Correlations between the Oil and Gas subsector operating cash flows and that of the GPCF, in first differences. The subsectors are weighted by the FTSE GEISAC index weights.

| Subsector | Correlation | *t*-Value |
|---|---|---|
| Exploration and Production | 0.52 | 2.18 * |
| Integrated Oil and Gas | 0.88 | 6.67 **** |
| Oil Equipment and Services | 0.10 | 0.37 |
| Pipelines | 0.21 | 0.69 |
| Renewable Energy Equipment | −0.59 | −1.95 |
| Alternative fuels | −0.09 | −0.23 |
| Oil and Gas ex Exploration and Production | 0.88 | 6.60 **** |

**** $p < 0.001$, * $p < 0.1$.

## 6. Robustness

We first checked for possible effects of heteroscedasticity by looking at the log rates of returns implied by a set of pseudo-asset values constructed from the cash flows in order to check the robustness of our results. Second, we redid the original analysis with different weighting schemes. Third, we considered the possible changes over time by studied rolling correlations. Finally, we looked for evidence of cross correlations at different leads and lags within our sample.

### 6.1. A Rough Check for Heteroscedasticity

Our use of first differences of nominal quantities might raise concerns regarding heteroscedasticity to the extent that inflation and/or real growth make the first differences grow over time. Although not affecting the consistency of our estimates, heteroscedasticity could bias our t statistics. We have reshaped our data, as follows, for a possible correction, despite the apparent absence of strong trends in our level data, as illustrated by the flattish shape of the respective curves in Figure 2.[9] First, we constructed pseudo-asset values for each industry and the GPCF for 2002, defined as the discounted sum of their respective cash flows for 2003–2018. For this purpose, we used a discount rate of 3.5%. The results were not sensitive to this choice. Subsequently, we constructed pseudo-asset values for the following years by adding each year's cash flow to the preceding year's pseudo-asset value. Finally, we constructed pseudo-log rates of return as 100 times the log difference between the annual pseudo-asset values. The correlation estimates for the pseudo-log rates of return, as shown in Table 10, show an even stronger correlation for the oil and gas industry than what we found on the straight first-difference data. Furthermore, the results in Table 10 also confirm the unique position of the oil and gas industry, although we now find significant correlations for basic materials and utilities. Perhaps needless to say, basic materials and utilities industries share some of the basic characteristics of oil and gas, as could be seen in Figure 2. In Section 6.4, below, we find some further reasons to keep an eye on these industries.

---

[9] This method was suggested to us by an anonymous referee.

**Table 10.** Correlations for pseudo-log returns between the GPFG and the GPCF, 2003–2018.

| Industry | Correlation | *t*-Value |
|---|---|---|
| Oil and Gas | 0.91 | 8.04 **** |
| Basic Materials | 0.65 | 3.23 *** |
| Industrials | 0.41 | 1.70 |
| Consumer Goods | 0.05 | 0.18 |
| Health Care | −0.23 | −0.88 |
| Consumer Services | 0.07 | 0.25 |
| Telecommunications | 0.24 | 0.92 |
| Utilities | 0.66 | 3.32 *** |
| Financials | 0.16 | 0.61 |
| Technology | −0.48 | −2.05 * |
| All industries | 0.43 | 1.76 |
| All industries ex Oil and Gas | 0.25 | 0.95 |

**** $p < 0.0001$, *** $p < 0.01$, * $p < 0.1$.

### 6.2. Alternative Weighting Schemes

We weighted our cash-flow observations by first adding up the cash flows for the companies in each industry and then weighting the sum for each industry according to the respective industry weights in the GPFG strategic index, as explained in Section 4. As our first alternative weighting scheme, we repeated this procedure, but with the FTSE GEISAC industry weights. The difference is mainly that the GPFG index overweights European companies, whereas the FTSE GEISAC does not. The results are reported in Table A1 in Appendix A. The differences from the ones in Table 8 above are trivial.

Our second alternative weighting scheme also used the FTSE GEISAC weights. However, we weighted the flows of each company individually instead of adding up the cash flows of all the companies in each industry. We then needed to rescale the company weights to make the implied industry weights equal the ones in the FTSE GEISAC index because of missing information for some companies. The results, as presented in Table A2 in the Appendix A, are also very similar to the ones in Table 8.

Our third alternative was to ignore weighting altogether and simply add up the cash flows as reported by each company within the ICB industries. Not unexpectedly, these results differed some more from the ones in Table 8. However, the strong correlation for the Oil and Gas industry remains.

For the Oil and Gas subsectors, we were unable to redo the results with the first of the three alternative schemes that are mentioned above, because we could not obtain the GPFG subsector weights for the respective years, only the ones for 2018 that are displayed in Table 2. However, we were able to repeat the analysis while using the second and third alternative weighting schemes. The results, as shown in Table A4 in Appendix A, are very similar to the ones presented in Table 9 above. The only difference of interest is that the correlation for the Exploration and Production now is significant on the 1 percent level; however, it remains lower than the one for Integrated Oil and Gas.

### 6.3. Rolling Estimates

The scope for studying changing correlations over time is limited, given the short length of our data sample. However, we have carried out seven-year rolling correlation estimates for the major industries, again in first differences. The results are displayed graphically in Figure 3. Interestingly, Health Care, and to some extent Telecommunications, show rather high correlations with the government's cash flow from petroleum activity in about the first half of the sample; but the corresponding correlations subsequently seem to collapse. Oil and Gas is the only major industry whose correlation is virtually unchanged during the entire period; and its correlation coefficient consistently lies around 0.8.

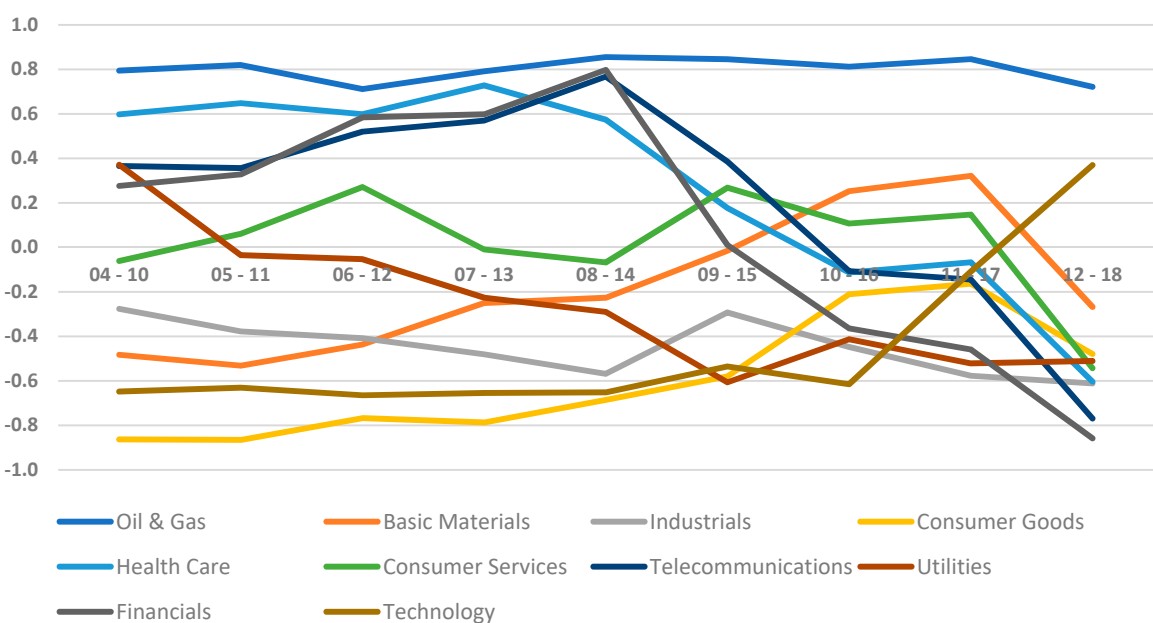

**Figure 3.** Rolling estimates of correlations.

*6.4. Cross Correlograms*

Our final robustness check concerns correlations at various leads and lags. Specifically, for each of the major industries, we computed correlations for the industry cash flow at time $t + k$ with the Norwegian government's petroleum cash flow at time $t$. The results are presented as a cross correlogram for each industry. Thus, the column, labeled "−3", denotes the correlation between the government's petroleum cash flow and the cash flow of the industry in question three years earlier, and so on.

In the results presented in Figure 4, one item stands out, namely, a correlation of 0.8 between the government's cash flow and the cash flow of the Basic Materials industry one year earlier. We interpret this finding as supportive of the notion that fossil fuels often move together with other materials. We note it as a possible argument for underweighting the shares in the Basic Materials industry as well as Oil and Gas. However, given the one-year lag, we recognize that this argument is much weaker than for Oil and Gas. We also find correlations above 0.6 for utilities, but at two-year leads and lags.

Furthermore, we constructed similar cross correlograms for the three subsectors of the Oil and Gas industry, for which we have data for the entire sample period. Figure 5 shows the results. There, we note the correlations of around 0.5 for Oil Equipment and Services at both the one-year lag and the one-year lead. These modestly large correlations may perhaps reflect the leads and lags in the industry itself, in that higher investment might presage higher government petroleum revenues and also that high government revenues (due, e.g., to higher prices) may signal higher future investments in the industry. However, the lack of contemporaneous correlation makes such a conclusion tenuous.

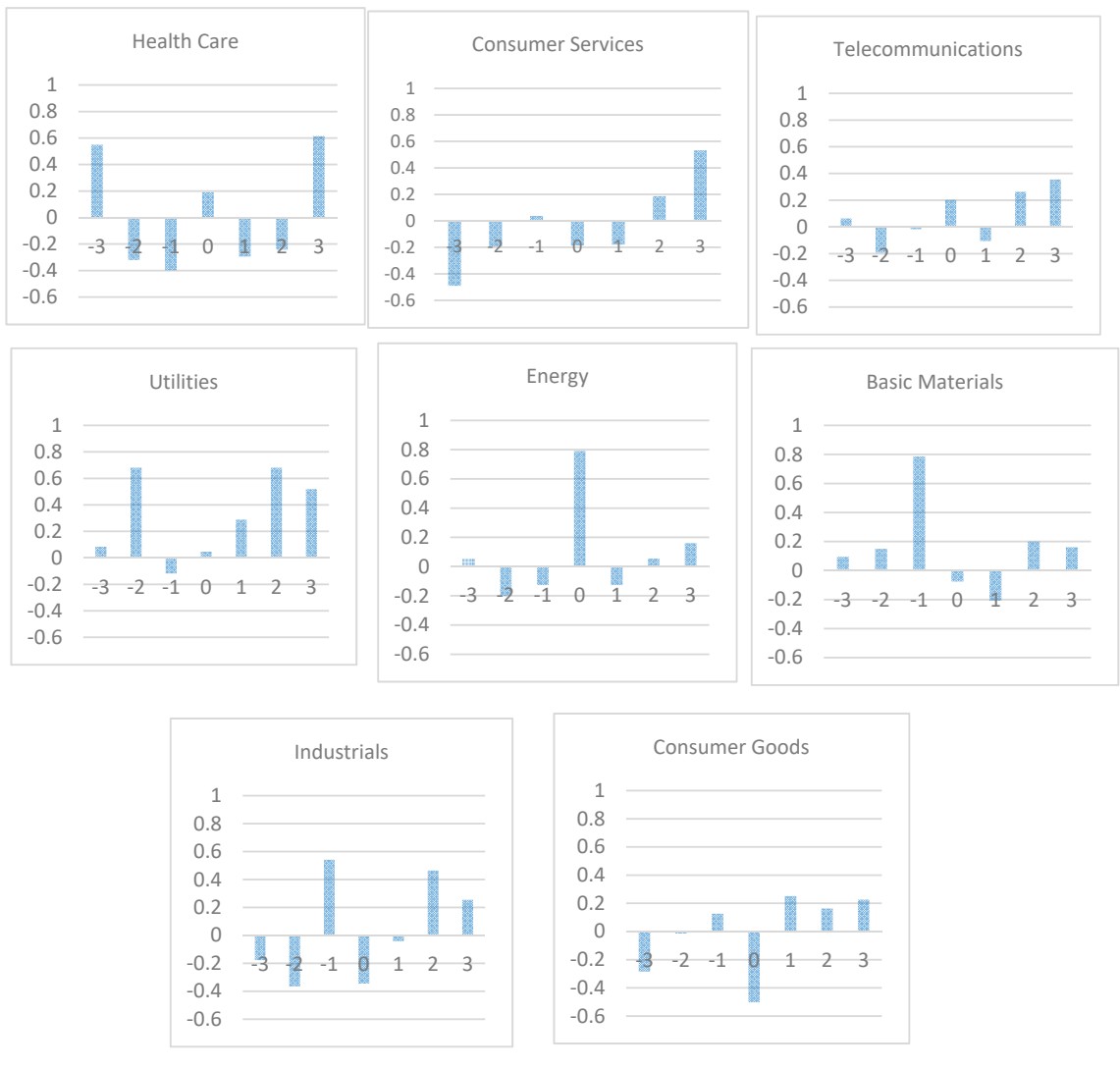

**Figure 4.** Cross correlograms for leads and lags of industry cash flows.

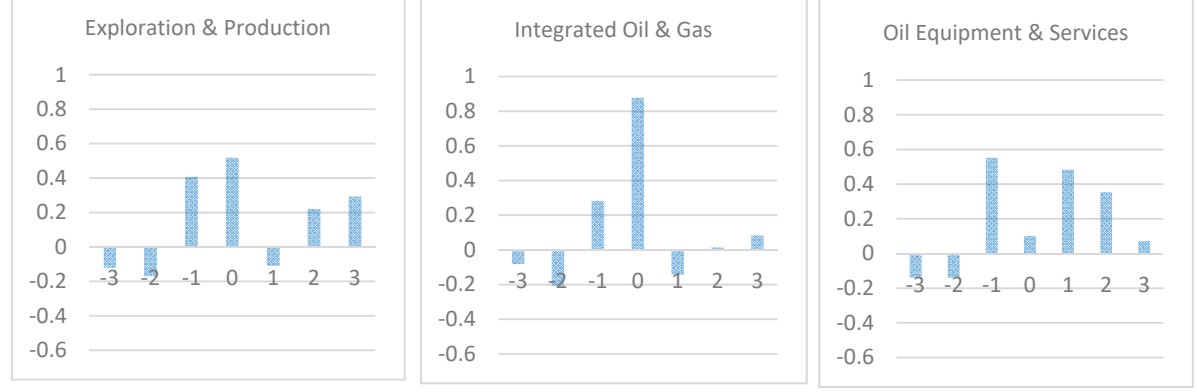

**Figure 5.** Cross correlograms for leads and lags of Oil and Gas subsector cash flows.

### 7. Conclusions

Ultimately, equities are all about cash flows, the right to current and future cash flows, with future cash flows being discounted at stochastic rates that may change over time. Thus, cash flows form the basis for equity prices. Investors, in turn, choose portfolios that are appropriately diversified, so as to minimize the risk given the rate of return. However, they also need to keep an eye on background risks to make sure that correlations between those risks and financial risks don't introduce unwanted concentration risks.

Assessing such risks often runs into problems regarding how to compare the different risks. This paper studies this problem for a government that regularly receives special revenues from the extraction of a non-renewable natural resource and, at the same time, holds a portfolio of equities that includes shares in the companies of that same industry. The comparison problem arises because the estimation of the correlation between asset prices in that industry and the government's value of the assets in the ground is infeasible because the value of the asset in the ground is unobservable. Some previous attempts to solve this problem have used the market price of the extracted resource as a proxy for the value of the asset in the ground. Others have sought to econometrically distinguish those movements in the equity prices for companies in the ground that are driven by changing expectations of cash flows from those that are due to time variations in discount factors. In this paper, we have bypassed this problem by instead directly comparing the government's cash flow from the resource to the aggregated cash flows of the global companies in the resource industry in question.

In particular, we have studied the correlations between the Norwegian government's petroleum cash flow with the cash flows of companies in the global oil and gas industry, as well as with companies in other major industries. We find a uniquely high correlation between the government's cash flow and the cash flow of companies in the global oil and gas industry. This correlation is unmatched, by a wide margin, by any other industry. Within oil and gas, we find the greatest correlations for the subsector containing integrated oil and gas companies and a somewhat weaker, yet significant, correlation for the subsector of oil and gas companies with upstream companies only.

For the Norwegian Government Pension Fund Global, we believe that our findings strongly support the underweighting, perhaps even shorting, of oil and gas shares in particular, and particular for the large, integrated oil and gas companies.

However, we believe that our methodology should be relevant to other governments as well whose revenues come from other kinds of natural resources, such as coal, iron ore, copper, bauxite, lithium, or—in the future—wind or sunshine. Further investigation of such cases would be an interesting extension of our work.

**Author Contributions:** Conceptualization, K.A.M.; methodology, K.A.M.; software, H.M.E. and M.E.H.; validation, K.A.M.; formal analysis, H.M.E. and M.E.H.; investigation, H.M.E. and M.E.H.; resources, H.M.E. and M.E.H.; data curation, H.M.E. and M.E.H.; writing—original draft preparation, H.M.E. and M.E.H.; writing—review and editing, K.A.M.; visualization, K.A.M.; supervision, K.A.M.; project administration, K.A.M.; funding acquisition, K.A.M. All authors have read and agreed to the published version of the manuscript.

**Funding:** K.A.M. acknowledges financial assistance from Finansmarkedsfondet, the financial markets research fund under The Research Council of Norway, Project no. 294398.

**Acknowledgments:** This paper is the revised version of a MSc thesis written at NTNU by H.M.E. and M.E.H. under K.A.M.'s supervision. The authors are indebted to two anonymous referees and to Halvor Hoddevik for suggesting the topic and for help with the data collection as well as the analytical methods.

**Conflicts of Interest:** The authors declare no conflict of interest.

## Appendix A

**Table A1.** Correlations between the operating cash flow of each industry and the GPCF. Industry cash flow weighted with FTSE GEISAC industry weights. All correlations on first-difference data 2004–2018.

| Industry | Correlation | *t*-Value |
|---|---|---|
| Oil and Gas | 0.80 | 4.81 *** |
| Basic Materials | −0.11 | −0.40 |
| Industrials | −0.30 | −1.12 |
| Consumer Goods | −0.52 | −2.20 * |
| Health Care | 0.13 | 0.48 |
| Consumer Services | −0.04 | −0.14 |
| Telecommunications | 0.02 | 0.05 |
| Utilities | 0.35 | 1.33 |
| Financials | −0.15 | −0.54 |
| Technology | −0.21 | −0.76 |
| All industries | −0.12 | −0.43 |
| All industries ex Oil and Gas | −0.25 | −0.93 |

*** $p < 0.01$, * $p < 0.1$.

**Table A2.** Correlations between the operating cash flow of each industry and the GPCF. Each company's cash flow weighted individually with the weights in the FTSE GAISAC index, scaled to correct for missing observations.

| Industry | Correlation | *t*-Value |
|---|---|---|
| Oil and Gas | 0.88 | 6.82 **** |
| Basic Materials | 0.10 | 0.38 |
| Industrials | 0.29 | 1.11 |
| Consumer Goods | −0.20 | −0.75 |
| Health Care | 0.44 | 1.77 |
| Consumer Services | 0.36 | 1.38 |
| Telecommunications | 0.25 | 0.91 |
| Utilities | 0.38 | 1.50 |
| Financials | −0.33 | −1.28 |
| Technology | −0.06 | −0.23 |
| All industries | 0.26 | 0.96 |
| All industries ex Oil and Gas | −0.17 | −0.63 |

**** $p < 0.0001$.

**Table A3.** Correlations between the government's petroleum cash flow and the unweighted operating cash flows of each ICB industry.

| Industry | Correlation | *t*-Value |
|---|---|---|
| Oil and Gas | 0.80 | 4.86 *** |
| Basic Materials | 0.20 | 0.72 |
| Industrials | −0.27 | −1.00 |
| Consumer Goods | −0.61 | −2.80 ** |
| Health Care | −0.27 | −1.01 |
| Consumer Services | −0.14 | −0.52 |
| Telecommunications | −0.46 | −1.87 * |
| Utilities | −0.02 | −0.07 |
| Financials | −0.01 | −0.03 |
| Technology | −0.13 | −0.46 |
| All industries | −0.04 | −0.15 |
| All industries ex Oil and Gas | −0.21 | −0.79 |

*** $p < 0.01$, ** $p < 0.05$, * $p < 0.1$.

**Table A4.** Correlations between the GPCF and the operating cash flows of the respective subsectors of Oil and Gas. Each company's cash flow weighted individually with the weights in the FTSE GAISAC index, scaled to correct for missing observations.

| Subsector | Correlation | *t*-Value |
|---|---|---|
| Exploration and Production | 0.74 | 3.94 *** |
| Integrated Oil and Gas | 0.89 | 7.11 *** |
| Oil Equipment and Services | 0.23 | 0.84 |
| Pipelines | 0.37 | 1.27 |
| Renewable Energy Equipment | −0.74 | −2.88 ** |
| Alternative fuels | 0.19 | −0.51 |
| Oil and Gas ex Exploration and Production | 0.89 | 7.16 *** |

*** $p < 0.01$, ** $p < 0.05$.

## References and Note

Baldwin, Christopher. 2012. *Sovereign Wealth Funds: The New Intersection of Money and Politics*. Oxford: Oxford University Press.

Benzoni, Luca, Pierre Collin-Dufresne, and Robert S. Goldstein. 2007. Portfolio choice over the life-cycle when the stock and labor markets are cointegrated. *Journal of Finance* 62: 2123–67. [CrossRef]

Bodie, Zvi, and Marie Brière. 2013. *Sovereign Wealth and Risk Management: A Framework for Optimal Asset Allocation of Sovereign Wealth*. Boston: Boston University.

Bodie, Zvi, Robert C. Merton, and William F. Samuelson. 1992. Labor supply flexibility and portfolio choice in a life cycle model. *Journal of Economic Dynamics and Control* 16: 427–49. [CrossRef]

Campbell, John Y. 1991. A variance decomposition for stock returns. *The Economic Journal* 101: 157–79. [CrossRef]

Campbell, John Y., and Robert J. Shiller. 1988. The Dividend-Price Ratio and Expectations of Future Dividends and Discount Factors. *The Review of Financial Studies* 1: 195–228. [CrossRef]

Campbell, John Y., Christopher Polk, and Tuomo Vuolteenaho. 2010. Growth or glamour? Fundamentals and systematic risk in stock returns. *Review of Financial Studies* 23: 305–44. [CrossRef]

Chambers, David, Elroy Dimson, and Aantti Ilmanen. 2012. The Norway Model. *The Journal of Portfolio Management* 38: 67–81. [CrossRef]

Cochrane, John H. 2014. A mean-variance benchmark for intertemporal portfolio theory. *Journal of Finance* 69: 1–49. [CrossRef]

Cohen, Randolph B., Christopher Polk, and Tuomo Vuolteenaho. 2009. The price is (almost) right. *Journal of Finance* 64: 2739–87. [CrossRef]

FTSE Russell. 2019. *Industry Classification Benchmark (ICB)*. Available online: https://www.ftserussell.com/data/industry-classification-benchmark-icb (accessed on 25 September 2019).

Heaton, John, and Deborah Lucas. 2000. Portfolio choice in the presence of background risk. *Economic Journal* 110: 1–26. [CrossRef]

Henriksen, Espen, and Jens Kværner. 2018. *Portfolio Choice with Non-Tradable Assets*. Oslo: BI Norwegian Business School.

Hoddevik, Halvor. 2018. Problemer i ekkokammeret ("Problems in the echo chamber"). *Dagens Næringsliv*, October 6.

Lintner, John. 1965. Security Prices, Risk, and Maximal Gains from Diversification. *Journal of Finance* 20: 587–615.

Norges Bank Investment Management. 2017a. Petroleum Wealth and Oil Price Exposure of Equity Sectors. Discussion note 04/2017.

Norges Bank Investment Management. 2017b. Strategy 2017-2019. Available online: https://www.nbim.no/en/publications/strategy-for-the-fund-management/strategy-plan-2017-2019/ (accessed on 28 May 2019).

Norges Bank Investment Management. 2018. *Investment strategy*. Available online: https://www.nbim.no/en/the-fund/how-we-invest/investment-strategy/ (accessed on 28 May 2019).

Norges Bank Investment Management. 2019. Holdings as at 31.12.2018. Available online: https://www.nbim.no/en/the-fund/holdings/holdings-as-at-31.12.2018/?fullsize=true (accessed on 28 May 2019).

Norwegian Ministry of Finance. 2001. Retningslinjer for den økonomiske politikken. In *Guidelines for Economic Policy*; St.meld. nr. 29 (2000–2001); Oslo: Norwegian Ministry of Finance, Not available in English.

Norwegian Ministry of Finance. 2015. *Finanspolitikk i en Oljeøkonomi—Praktisering av Handlingsregelen*; NOU 2015:9; Oslo: Norwegian Ministry of Finance, (English translation of Chapter 1: *Fiscal Policy in an Oil Economy—The Application of the Fiscal Rule*).

Norwegian Ministry of Finance. 2016a. *Aksjeandelen i Statens Pensjonsfond Utland*; NOU 2016:20; Oslo: Norwegian Ministry of Finance, (*Official Norwegian Report—The Equity Share of the Government Pension Fund Global*. Unofficial translation of chapter 1).

Norwegian Ministry of Finance. 2016b. *The Management of the Government Pension fund in 2016*; Meld. St. 26, (2016–2017); Oslo: Norwegian Ministry of Finance, Report to the Storting (white paper).

Norwegian Ministry of Finance. 2018. Energiaksjer i Statens Pensjonsfond Utland. In *Energy Shares Int the Government Pension Fund Global*; NOU 2018:12; Oslo: Norwegian Ministry of Finance, Not available in English.

Norwegian Ministry of Finance. 2019. *The Report on Energy Stocks in the Government Pension Fund Global*; Oslo: Norwegian Ministry of Finance.

Pettit, R. Richardson, and Randolph Westerfield. 1972. A model of capital asset risk. *Journal of Financial and Quantitative Analysis* 7: 1649–68. [CrossRef]

Scherer, Bernhard. 2019a. *A Note on Portfolio Choice for Sovereign Wealth Funds*. Nice: EDHEC-Risk Institute.

Scherer, Bernhard. 2019b. *Portfolio Choice for Oil-Based Sovereign Wealth Funds*. Nice: EDHEC-Risk Institute.

Sharpe, William F. 1964. Capital Asset Prices: A Theory of Market Equilibrium under Conditions of Risk. *Journal of Finance* 19: 425–42.

van den Bremer, Ton, Frederick van der Ploeg, and Samuel Willis. 2016. The Elephant in the Ground: Managing Oil and Sovereign Wealth. *European Economic Review* 82: 113–31. [CrossRef]

Viceira, Luis M. 2001. Optimal Portfolio Choice for Long-Horizon Investors with Nontradable Labor Income. *Journal of Finance* 56: 433–70. [CrossRef]

