# Peer review of "Portfolio Choice for a Resource-Based Sovereign Wealth Fund: An Analysis of Cash Flows"

_ijfs, doi:10.3390/ijfs8010014_

Round 1

Reviewer 1 Report

Please see attached pdf file with comments for authors.

Author Response

Response to Referee #1

We want to thank you for your incisive comments as well as your most constructive suggestions. To your three comments we have the following responses:

  1. Thank you very much for these references to literature of which we were not aware. We have included the references, along with short descriptions of their respective contributions, on lines 97 – 103 in our revised manuscript.
  2. We appreciate the clarification of the different roles of dividends and company cash flows. In response, we have sought to make the distinction clear to the reader while also emphasizing the underlying role of company cash flows in creating values for stock holders. We also emphasize that the Norwegian government’s petroleum revenues are more similar to company cash flows than to dividends. We seek to clarify this issue in lines 104 – 113 in our revised manuscript. This paragraph replaces the remarks on this issue in our original submission. Furthermore, the new footnote 8 clarifies that the present-value formula in line 424 properly applies only to dividends. As a final detail, in line 74, we have the wording from “expectations of future cash flow” rather than “expectations of company cash flow.”
  3. We share your wish to look at correlations further back in time and made a serious effort in this direction. Unfortunately, however, data availability becomes a serious obstacle. The problem seems to be that the providers of equity indices did not identify oil and gas as an industry until the mid-2000s. In fact, some of the data in the early part of our sample had to be assembled manually. To obtain reliable data further back in time, we would effectively have had to manually construct our own index. That would not only have been prohibitably time consuming, but also likely to result in biased data because we would have had to rely mainly on a few, major corporations; and even those would have been complicated by mergers, acquisitions, and company births and deaths. Finally, because the Norwegian government’s petroleum revenues were much smaller before the turn of the century, we find it likely that their movements then were more influenced by idiosyncratic factors that could bias the correlations. We have thus decided to ask for your forbearance with not taking on this challenge for the present paper. However, we have discussed the issue in lines 113 – 122, along with the new footnote 2 in the revised manuscript.

Reviewer 2 Report

The authors investigate whether reducing oil and gas investments in the Norwegian sovereign wealth fund (Norwegian Govermental Pension Fund Global, NGPFG) could benefit diversification of the fund. The owner, the Norwegian government, receives substantial income from its oil and gas sector. If some of its assets are highly correlated with this sector, the government may benefit from reducing exposure to this sector.

The authors assume that if the cash flow in the recent history of two investments is highly correlated, then it is likely that the value of these investments also is correlated. The implicit assumption is that the historical correlations will last. Hence, if the future cash flow of the government revenue is highly correlated with the revenue of oil and gas companies, then the value of this cash flow will be highly correlated too, and the government could benefit from diversification.

The authors therefore compare the cash flows of the government with those of private companies, including those in the oil and gas company. In spite of a very small sample, they find a very significant positive correlation between the government cash flow and that of oil and gas companies. This correlation is not present for the other industries. This correlation survives a series of robustness checks.

My assessment of the paper:

I think the authors make a convincing argument that governments with high revenues from certain industries (usually oil and gas) and with sovereign wealth funds, should divert their SWF investments away from the industry they depend the most on. It is also good reason to think that cash flows measure correlation of investment value more precise than alternatives.

The result seems to apply to a quite narrow set of cases. The advice of diversification is valid only for a handful of governments in the world today. However, the impact of policy changes here would nevertheless be very noticeable in financial markets because of the size of these funds and the importance of the oil and gas industry in the world’s capital markets. I therefore nevertheless think the paper makes important contribution to the financial literature.

The statistical methods used are not very advanced, which presumably has to do with a very small sample. It is very limited what you can do with such a small dataset before the degrees of freedoms are exhausted. Obtaining better data for this particular analysis might not be possible, though.

The results rely on correlation of first difference in levels. I think the authors would have reduced the likelihood of heteroscedasticity by converting the cash flow to a log return series. This could for example been achieved by calculating the present value of the cash flows in 2002 using some common discount rate, and then obtaining a proxy for the investment value each year by adding the cash flows. At least this would have been a useful robustness check.

The purpose of this would not be to estimate the true value of investments, but to obtain a normalized return series less likely to contain heteroscedasticity problems.

I also think that with such a small dataset, the authors should put the full dataset in a table, for example by adding the government cash flow to Table 5.

Author Response

Response to Referee #2

Thank you very much for your thorough review of our paper as well as your constructive comments. Our responses to your two suggestions are as follows:

  1. Although the data display in our figure 2 does not suggest serious heteroscedasticity from the use of first differences of nominal data, we have carried out the procedure that you propose. The results are presented as the new subsection 6.1 of the Robustness section. We have explained the procedure in lines 536 – 556 and present the results as the new Table 10. The results were slightly surprising. On the one hand, the uniqueness of the Oil and Gas industry was strengthened rather than weakened. On the other hand, we now also find significant, albeit much lower, correlations for Basic Materials and Utilities. Perhaps needless to say, the latter two industries are similar to Oil and Gas by being either extractive or engaged in energy-related activities.
  2. We have taken you up on your suggestion to add the Norwegian government’s petroleum cash flows as an additional column in Table 5. We hope the editors are still able to fit the table onto the page.